# Carbon sequestration characteristics of two plantation forest ecosystems with different lithologies of karst

Yuanyuan Li[1,2☯], Huiwen Xiang[3☯], Zongsheng Huang [1,4]*, Yuanbo Zhang[4], Jun Zou[1], Yuhong Fu[5], Changjiang Qian[5]

**1** College of Architecture and Urban Planning, Guizhou University, Guiyang, China, **2** Guizhou Botanical Garden, Guiyang, China, **3** School of Architecture and Art, Central South University, Changsha, China, **4** Guiyang Institute of Humanities and Technology, Guiyang, China, **5** School of Biological Sciences, Guizhou Education University, Guiyang, China

☯ These authors contributed equally to this work.
* hzsxjh@126.com

**Data Availability Statement:** Biomass data belongs to the sixth forest resource inventory data of Guizhou Forestry Bureau, which is unpublished by government departments. The name of the

## Abstract

In karst regions, the majority of studies have focused on ecosystem carbon sequestration in the same lithology, but no studies in different lithologies. In this study, actual measurements were used to reveal carbon sequestration characteristics of two plantation forest ecosystems (*Bodinieri cinnamon* and *Cupressus funebris*) with different lithologies of karst. The results showed that the tree layer showed the highest vegetation biomass, carbon content, carbon density, and ratio of aboveground biomass to belowground biomass. The carbon density of *B. cinnamon* plantation and *C. funebris* plantation was high in dolomite and in limestone respectively. The soil quality and carbon density of bare ground and plantation varied across different lithologies. The carbon density of various ecosystem components was in the order of vegetation>soil>litterfall. The carbon density and net carbon density of plantation varied across different lithologies. In *B. cinnamon* plantation, the carbon sequestration rate of vegetation and ecosystem was high in dolomite, moderate in limestone, and low in dolomitic sandstone. In *Cupressus funebris* plantation, the carbon sequestration rate was in the order of limestone>dolomite>dolomitic sandstone. These findings revealed that lithology is an important factor affecting ecosystem carbon pools, and plantation ecosystems have low biomass and low carbon density in karst areas.

## 1 Introduction

Global warming is an unavoidable problem. Scientists are actively exploring carbon peaking and carbon neutral strategies. To achieve carbon peaking and carbon neutrality, it is inevitable to identify the carbon sequestration rate of various ecosystems and their uncertainty, stability, and sustainability. However, these characteristics remain unclear for many ecosystems, particularly the karst regions, due mainly to high landscape heterogeneity, evident dualistic structure, small environmental capacity, discontinuous and shallow soil layer, exposed rocks, poor water storage, and weak resistance to disturbance [1, 2]. These challenges further complicate

organization that imposes restrictions on the data is Guizhou Forestry Bureau, and the contact email is: 527690165@qq.com.

**Funding:** This research was funded by National Natural Science Foundation of China (51868008, 51978187, and 31560187).The funders had no role in study design, data collection and analysis, decision to publish, or preparation of the manuscript.

**Competing interests:** The authors have declared that no competing interests exist.

the assessment of carbon sinks in the ecosystems of karst regions, which account for approximately 15% of the global land area and provide drinking water to nearly one-fourth of the global population [3, 4]. Therefore, the evaluation of carbon sequestration in karst ecosystems is of great theoretical and practical significance to assess carbon sinks in karst forests, establish carbon sink forests, and implement carbon neutral strategies.

Previous studies related to carbon sinks in the forest ecosystems of karst regions mainly focused on ecosystem carbon storage and its distribution in plantation forests of different ages [5–7]; carbon sequestration in urban forests [8, 9]; ecosystem carbon sequestration in the natural recovery of degraded karst forest vegetation [10]; carbon fluxes at different stages of the secondary succession of karst grasslands [6]; and carbon sinks of ecosystem components, including vegetation [11], soil [12–17], and litterfall [18]; however, there have been no studies on carbon sequestration in plantation forest ecosystems under different lithologies in karst regions. In karst regions, lithology is an important factor driving differences in plant habitats [2, 19], and the physicochemical properties and fertility of soil produced through weathering also vary under different lithologies, thus producing a greater impact on plant physiological and ecological functions [20, 21]. For instance, karst regions in the Guizhou Province of China, which is at the center of the karst regions of southern China, account for 25.8% of the total karst area in the country, with exposed carbonate rocks covering an area of $13 \times 10^4$ km$^2$. The Guizhou Province is home to the largest and the most developed tropical and subtropical karst in the world, including China [22]. According to the distribution of carbonate rock fabric [23], there are seven types of lithologies. Among these, limestone, dolomite, and dolomitic sandstone account for 42.73%, 17.31%, and 6.59%, respectively, of the total area of carbonate rocks in the Guizhou Province. *Cupressus funebris* and *Bodinieri cinnamon* are the dominant plant species growing in soils developed from these lithologies. Therefore, in the present study, we selected *C. funebris* and *B. cinnamon* planted on dolomite, limestone, and dolomitic sandstone in the Guizhou Province and obtained field measurements for studying the carbon sequestration of these two plantation ecosystems to reveal the composition of and changes in plantation ecosystem biomass and carbon density, soil carbon pool characteristics, litterfall carbon pool characteristics, and ecosystem carbon sequestration capacity under different lithologies in karst regions. Our findings can serve as the foundation for carbon sink assessment, ecosystem service function evaluation, carbon sink forest establishment, and vegetation restoration in karst regions.

## 2 Materials and methods

### 2.1 Overview of the study area

The Guizhou Province is located at the eastern end of the Yunnan–Guizhou Plateau in southwestern China (24˚37′ to 29˚13′N, 103˚36′ to 109˚36′E), with terrain descending in steps from the northwest to southeast and an average elevation of 1,100 m. In this region, karst landforms are typically developed and widely distributed, with distinct morphological types and obvious geographical differentiation, constituting a special karst ecosystem. Therefore, the Guizhou Province is an ideal area for studying karst forest ecosystem restoration. The province has monsoon-influenced humid subtropical climate, and owing to the mountainous terrain, the climate is complex and diverse. The western part belongs to the warm temperate zone, the southern and northwestern parts belongs to the southern subtropical zone, and the rest of the region gradually transitions from the central subtropical zone to the northern subtropical zone, with the raising of Miaoling, Wuling Mountains, and Dalou Mountains. The mean annual temperature and precipitation are 10–18˚C and 1,100–1,500 mm, respectively. The relative humidity is over 70%, and the annual sunshine hours are 1,300. The frost-free period is

270 days. The zonal soil belongs to the middle subtropical evergreen broad-leaved forest red soil—the yellow soil zone. Furthermore, the Guizhou Province harbors complex and diverse vegetation, which can be divided into five categories, including coniferous forests, broad-leaved forests, bamboo forests, scrub forests and grasslands, and swamp and aquatic vegetation.

## 2.2 Research methodology

**2.2.1 Field selection.** As mentioned above, limestone, dolomite, and dolomitic sandstone account for respectively 42.73%, 17.31%, and 6.59% of the total carbonate rocks in the Guizhou Province and represent the main lithologies in the karst area of this province. The present study selected fully developed regions of these three lithologies under the same condition of other environmental factors as the research field. The data were derived from the 2017 Guizhou Province Forest Resources Class II Survey and other field investigations. Common tree species planted on each lithology, namely *C. funebris* and *B. cinnamon*, were selected. *Cupressus funebris* and *B. cinnamon* forests were further grouped by age (same or similar), as described by Meng (2019) [24]. Experimental fields measuring 20 m × 20 m were set up in *C. funebris* and *B. cinnamon* forests of the same or similar age and on bare land with the three lithologies. Three replicates were set up for each forest and bare land, totaling 27 sample plots. The plots were selected based on the classification parameters of the degree of rocky desertification [9]. Before vegetation restoration, all plots were located in areas with intense rocky desertification. Here, vegetation carbon storage on the non-forested bare land was neglected because there were only a few herbaceous plants, and its soil carbon stock was used as the control for the forested ecosystem. Table 1 summarizes the details of sampling plots.

**2.2.2 Vegetation survey and sample collection and processing.** Vegetation survey: In the present study, the conventional community survey method was adopted [25]. In each sample plot, the following parameters were examined: (1) trees: species, number of plants, clear bole height, height, diameter at breast height (DBH; for trees taller than 3 cm, each tree was checked for size), crown width, cover, and frequency; (2) shrubs and herbs: species, number of plants, cover, and frequency; and (3) habitat factors: elevation, slope, slope direction, soil type, surrounding environment, and human disturbance status.

Sample collection and measurement: The biomass (kg) of herbaceous, shrub, and standard tree of the tree layers of the forest was measured using the harvesting method. The biomass (kg) of the tree layers of the forest was measured using the correlated growth method [25, 26]. The specific method is described below [10].

Biomass (kg) was surveyed and collected from 27 sample plots from August to October 2018. The height (m) of the herbaceous layer was ≤1.5 m (i.e., young trees and shrubs with height (m) of ≤1.5 m were grouped into the herbaceous layer). The height (m) of the shrub layer was >1.5 m and DBH (cm) was <3 cm. The DBH (cm) of the tree layer (woody plants) was ≥3 cm. In the herbaceous layer, nine quadrats measuring 1 m × 1 m were randomly selected from each sample plot. In the shrub layer, a single quadrat measuring 5 m × 5 m was randomly selected. In the tree layer, all trees with DBH (cm) of ≥3 cm in the sample plots were counted for each diameter class and average diameter class. Standard trees were counted according to DBH(cm), and three standard trees were selected from each plot. The aboveground and belowground parts of the selected herbs and shrubs from the small sample plots and of the standard trees of the tree layer were collected using the harvesting method. Leaves, branches, stems, and roots of the standard trees of the tree layer were harvested using microblasting or large digging machines. Finally, total fresh weight of each sample and fresh weight of the aboveground and belowground parts were measured. Some samples were dried to a

**Table 1. Basic overview of the sample sites.**

| Lithology | Geographical area | General information of the study area | Sample type | Forest age group | Dominant species age | Basic characteristics of the samples |
|---|---|---|---|---|---|---|
| Dolomite | 26°49′44″ to 26°51′37″N | The study site is located on the right bank of the Xiuwen River in Shaxi Village, Longchang Town, Xiuwen County, Guiyang City, Guizhou Province, at an altitude of 1,100–1,500 m. The study site belongs to the subtropical monsoon climate zone. Mean annual temperature is 16.5°C, with active cumulative temperature of 4,097.40°C. Mean annual rainfall is 1,100 mm, and there is 1,359.4 h of sunshine throughout the year. The soil layer is shallow and has a high gravel content, with yellow loam and black limestone soils developed mainly from dolomitic tuff. The existing vegetation includes diverse secondary trees and scrub forests, shrub forests, and vine scrub forests, with 80% vegetation cover. | *Bodinieri cinnamon* forest | Young | 16-year-old | Elevation 1,358 m, east facing, slope 25°, limestone soil, pure plantation forest. The community structure is simple. The tree layer is *Bodinieri cinnamon*, with height of 4–8 m and coverage of 80%. |
| | 106°35′29″ to 106°33′21″E | | *Cupressus funebris* forest | Young | 16-year-old | Elevation 1,414 m, northeast facing, slope 40°, black limestone soil, pure plantation forest. The community structure is simple. The tree layer is *Cupressus funebris*, with height of 5–8 m and coverage of 30%. |
| Dolomitic sandstone | 26°33′53″ to 26°34′40″N | The study site is located in Kaitang Town, Kaili City, southeastern Guizhou Province, at an altitude of 523–1,233 m. The site belongs to the central subtropical warm and humid monsoon climate zone. Mean annual temperature is 13.6–16.2°C. Mean temperature in January is 2.6–5.2°C. Mean annual sunshine hours are 1,289.1, and mean annual precipitation is 1,240.4 mm. Average annual relative humidity is 78%, and the frost-free period is 288 days. Limestone and yellow-brown loamy soils are predominant. The existing vegetation is mainly natural secondary and planted forests. | *Bodinieri cinnamon* forest | Young | 13-year-old | Elevation 615 m, east facing, slope 38°, yellow limestone soil, pure plantation forest. The community structure is relatively simple, with height of 2.5–6 m and coverage of 30%. |
| | 107°59′08″ to 107°56′48″E | | *Cupressus funebris* forest | Middle-aged | 38-year-old | Elevation 626 m, south facing, slope 30°, limestone soil, pure plantation forest. The community structure is complete. The tree layer is the most developed, with height of 5–8 m and coverage of 75%. |
| Limestone | 25°46′52″ to 25°55′59″N | The study site is located in Yongning and Puli towns, Guanling County, Anshun City, Guizhou Province, at an altitude of 800–1,500 meters. Mean average temperature is 16.2°C, with the maximum mean temperature of 16.9°C and the minimum temperature of 15.4°C. There is abundant rainfall, with annual precipitation of 1,205.1–1,656.8 mm. The site belongs to humid mid-subtropical monsoon climate zone. There are four distinct seasons, with rain and heat in the same season. Limestone and yellow-brown loamy soils are predominant. The existing vegetation is mainly natural secondary and plantation forests. | *Bodinieri cinnamon* forest | Young | 16-year-old | Elevation 1,432 m, southeast facing, slope 15°, limestone soil, pure plantation forest. The community structure is homogeneous, with height of 2.5–5 m. The tree layer cover is 50%. |
| | 105°34′09″ to 105°28′03″E | | *Cupressus funebris* forest | Young | 16-year-old | Elevation 1,545 m, northeast facing, slope 45°, black limestone soil, pure plantation forest. The community structure is simple, with a few shrubs. The tree layer is *Cupressus funebris*, with height of 4–8 m and coverage of 40%. |

constant weight in an oven at 80°C. The ratio of dry mass to fresh mass (kg) was separately calculated for each part. Then, the sample dry mass (kg) was calculated, and the biomass (t·hm$^{-2}$) per hectare of each sample in every sample plot was estimated. The biomass of each harvested organ of the standard trees was subjected to regression analysis using, and the following regression equation was obtained [10, 27–30]:

$$W = a(D^2H)^b,$$
(1)

where $W$ is the biomass of each part of the plant; $D$ is diameter at breast height; $H$ is tree height; and a and b are constants.

Finally, the biomass of the tree layer was calculated according to the regression equation.

**2.2.3 Measurement of vegetation organic carbon indicators.** The harvested samples were crushed three times using the quadratic method, sieved using a 100 mesh, sampled, and dried to a constant weight in an oven at 80°C. The carbon content ($g·kg^{-1}$) was determined using the $K_2Cr_2O_7–H_2SO_4$ external heating method [31,32]. Three replicates of tree, shrub, and herb samples were analyzed for each lithology. The weighted average carbon content ($g·kg^{-1}$) was calculated as the product of the carbon content of each part (or organ) and its weighting factor, that is, the ratio of the biomass of each part (or organ) to the overall biomass [10]. Carbon storage (g) was calculated as the product of the carbon content ($g·kg^{-1}$) and biomass (kg), and convert carbon storage (t) to tons in every sample plot. Carbon density ($t·hm^{-2}$) was calculated as the carbon storage divided by the area of the sample plot.

**2.2.4 Determination of organic carbon in the litterfall accumulation.** A square sampling frame measuring 1 m² was prepared using a thin iron strip (1 mm thick and 10 cm wide), and 30 small randomly selected quadrats in each plot were sampled for the existing stock of litterfall. The square sampling frame was placed horizontally during sampling, and the existing stock of litterfall within the area of the frame, projected vertically in the sample plot, was collected as described elsewhere [18]. The litterfall was collected, brought to the laboratory, dried to a constant weight in an oven at 80°C, and weighed. The litterfall accumulation was calculated based on dry weight. In addition, some dried samples were used to measure carbon content. Carbon density was calculated as the product of the litter fall accumulation per unit area ($t·hm^{-2}$) of the quadrat and the mean carbon content of the litterfall in the corresponding quadrat [25, 33].

**2.2.5 Determination of soil organic carbon content.** In the present study, soil organic carbon content, organic carbon density, and organic carbon storage were measured using the soil volume actual measurement method in two layers (0–20 cm and >20 cm), as described elsewhere [12]. Following collection, the samples were placed into sealed plastic bags, brought to the laboratory, air dried, and subjected to soil organic carbon measurement. To determine bulk weight, samples were collected with a ring knife sampler [15]. Organic carbon content was determined using the $K_2Cr_2O_7–H_2SO_4$ external heating method [31]. Gravel content was determined by passing the sample through a 2 mm soil sieve [15]. Soil organic carbon storage (400 m² sample plots) was calculated using the following equation:

$$CS_i(t) = C_i V_i D_i (1-G_i) \times 10^{-6}, \tag{2}$$

where $i$ represents the two soil layers (0–20 and >20 cm); $CS_i$ is the organic carbon storage of the corresponding soil layer; $C_i$ is the soil organic carbon content of the corresponding soil layer (%); $D_i$ is the bulk weight of the corresponding soil layer ($g·m^{-3}$); $G_i$ is the gravel content (>2 mm) of the corresponding soil layer (%); and $V_i$ is the measured soil volume in the corresponding soil layer (m³). Carbon density was calculated as the carbon storage divided by the sample area.

**2.2.6 Calculation of indicators related to organic carbon in ecosystems.** Organic carbon storage in the ecosystem was calculated as the sum of carbon storage in vegetation, litterfall, and soil in the sample plots. Organic carbon density in the ecosystem was calculated as the ratio of organic carbon storage to sample plot area. Net organic carbon density was calculated as the difference between organic carbon density in the ecosystem and bare ground. The carbon sequestration rate of vegetation was calculated as the organic carbon density of vegetation divided by the recovery time. The carbon sequestration rate of the ecosystem was calculated as

the difference between the organic carbon density of ecosystem and that of bare ground divided by the recovery time.

## 2.3 Data processing

The data were statistically analyzed using Excel and SPSS 17.0. One-way ANOVA and the least significant difference (LSD) test were used for comparisons of different datasets. The significance level was set at $\alpha = 0.05$.

## 3 Results

### 3.1 Vegetation biomass of the two plantation under different lithologies in karst regions

**3.1.1 Regression models for relative growth relationships of vegetation tree layers under different lithologies.** As shown in Table 2, the relationship of the biomass of each part of the tree ($W$) with DBH ($D$) and tree height ($H$) could fit well to the equation $W = a$

**Table 2. Relationship of the biomass of each part (kg) ($W$) of the dominant species with diameter at breast height (cm) ($D$) and tree height (m) ($H$) under different lithologies.**

| Lithology | Species | Part | Equation | $R^2$ | $P$ |
|---|---|---|---|---|---|
| Dolomite | *Cupressus funebris* | Leaf | $W = 0.17701 \, (D^2H)^{0.496}$ | 0.693 | <0.05 |
| | | Branch | $W = 0.12942 \, (D^2H)^{0.530}$ | 0.446 | <0.05 |
| | | Stem | $W = 0.21577 \, (D^2H)^{0.582}$ | 0.707 | <0.05 |
| | | Root | $W = 0.14521 \, (D^2H)^{0.525}$ | 0.787 | <0.05 |
| | | Total (statistics) | $W = 0.73451 \, (D^2H)^{0.490}$ | 0.718 | <0.05 |
| | *Bodinieri cinnamon* | Leaf | $W = 0.00723 \, (D^2H)^{0.914}$ | 0.619 | <0.05 |
| | | Branch | $W = 0.00094 \, (D^2H)^{1.328}$ | 0.761 | <0.05 |
| | | Stem | $W = 0.01422 \, (D^2H)^{0.965}$ | 0.932 | <0.01 |
| | | Root | $W = 0.00456 \, (D^2H)^{1.017}$ | 0.828 | <0.01 |
| | | Total (statistics) | $W = 0.02070 \, (D^2H)^{1.051}$ | 0.881 | <0.01 |
| Dolomitic sandstone | *Cupressus funebris* | Leaf | $W = 0.10209 \, (D^2H)^{0.674}$ | 0.962 | <0.01 |
| | | Branch | $W = 0.02032 \, (D^2H)^{0.925}$ | 0.954 | <0.01 |
| | | Stem | $W = 0.06194 \, (D^2H)^{0.875}$ | 0.988 | <0.01 |
| | | Root | $W = 0.03573 \, (D^2H)^{0.850}$ | 0.961 | <0.01 |
| | | Total (statistics) | $W = 0.18880 \, (D^2H)^{0.840}$ | 0.980 | <0.01 |
| | *Bodinieri cinnamon* | Leaf | $W = 0.04581 \, (D^2H)^{0.555}$ | 0.916 | <0.01 |
| | | Branch | $W = 0.07621 \, (D^2H)^{0.565}$ | 0.912 | <0.01 |
| | | Stem | $W = 0.10691 \, (D^2H)^{0.613}$ | 0.902 | <0.01 |
| | | Root | $W = 0.16672 \, (D^2H)^{0.388}$ | 0.993 | <0.01 |
| | | Leaf | $W = 0.36728 \, (D^2H)^{0.546}$ | 0.947 | <0.01 |
| Limestone | *Cupressus funebris* | Leaf | $W = 0.07980 \, (D^2H)^{0.654}$ | 0.417 | <0.05 |
| | | Branch | $W = 0.13646 \, (D^2H)^{0.573}$ | 0.763 | <0.05 |
| | | Stem | $W = 0.10209 \, (D2H)^{0.745}$ | 0.877 | <0.01 |
| | | Root | $W = 0.09036 \, (D^2H)^{0.614}$ | 0.871 | <0.01 |
| | | Total (statistics) | $W = 0.39902 \, (D^2H)^{0.658}$ | 0.823 | <0.05 |
| | *Bodinieri cinnamon* | Leaf | $W = 0.07396 \, (D^2H)^{0.502}$ | 0.417 | <0.05 |
| | | Branch | $W = 0.00865 \, (D^2H)^{1.121}$ | 0.532 | <0.05 |
| | | Stem | $W = 0.03412 \, (D^2H)^{0.889}$ | 0.89 | <0.01 |
| | | Root | $W = 0.09036 \, (D^2H)^{0.614}$ | 0.871 | <0.01 |
| | | Total (statistics) | $W = 0.30620 \, (D^2H)^{0.661}$ | 0.912 | <0.01 |

$(D^2H)^b$. Correlations with this equation reached the level of significance at $p < 0.05$. Dolomitic sandstone showed the optimal goodness-of-fit, followed by limestone, while dolomite showed the worst goodness-of-fit. These results indicate that plant growth was more consistent on dolomitic sandstone and more irregular on dolomite. Hence, dolomitic sandstone habitat is relatively consistent, followed by limestone habitat, while the dolomite habitat is complex and variable.

**3.1.2 Vegetation biomass composition of the two plantation species under different lithologies.**   As shown in Table 3, the overall vegetation biomass of B. cinnamon of the same or similar age was high in dolomite habitat, moderate in limestone habitat, and low in dolomitic sandstone habitat. Meanwhile, the overall vegetation biomass of *C. funebris* of the same age was higher in limestone than in dolomite habitat, indicating differences in species adaptability to different lithologies. The ratio of biomass of each layer to the vegetation biomass was high in the tree layer and low in the grass and shrub layer. The percentage of tree layer biomass to vegetation biomass was as follows: *B. cinnamon* in dolomite = 85.01%, *C. funebris* in dolomite = 76.37%, *B. cinnamon* in dolomitic sandstone = 50.08%, *C. funebris* in dolomitic sandstone = 88.25%, *B. cinnamon* in limestone = 71.10%, and *C. funebris* in limestone = 79.51%. All values were above 50%, indicating that vegetation biomass was dominated by the tree layer. At the same age, the biomass of *B. cinnamon* forest was greater than that of *C. funebris* forest in dolomite habitat, while the opposite result was noted in limestone habitat, indicating that *B. cinnamon* is highly adapted to dolomite, whereas *C. funebris* is highly adapted to limestone.

**3.1.3 Ratio of aboveground biomass to belowground biomass of the two plantation species under different lithologies.**   As shown in Table 4, the overall ratio of aboveground biomass to belowground biomass was in the order of tree layer>shrub layer>grass layer, indicating that grass and shrubs show a greater adaptability to karst habitats. The aboveground biomass to belowground biomass ratio of trees and vegetation was high in dolomite, moderate in dolomitic sandstone, and low in limestone. Therefore, dolomite has better quality of habitats. The ratio of aboveground biomass to belowground biomass of *C. funebris* in the tree layer on limestone was the highest at 3.78. The plantation plot was a demonstration site of *C. funebris* blast plantation in a rocky desertified habitat. In this type of plantation restoration, a planting hole is created via micro-blasting on the rock, and the planting soil is collected from thin soil on the bare ground. The planting hole is small, and the surrounding rocks restricts root growth. Therefore, the belowground rocky habitat has a greater influence on root growth, and the belowground habitat itself is influenced by lithology. These results indicate the selectivity of species to lithology, which is a challenge during plantation restoration.

## 3.2 Carbon pool characteristics of the two plantation forests under different lithologies in karst region

**3.2.1 Carbon content of vegetation in 2 plantation forests under different lithologies.** As shown in Table 5, the carbon content of vegetation was in the order of tree layer>shrub layer>herb layer, and the carbon content of each tree part was in the order of stem>branches>leaves>roots (tree layer). There was no significant difference between the aboveground and belowground carbon content of the shrub layer, indicating that lithology did not have a great impact on carbon content in forests. The weighted average carbon content of the stem and root of *C. funebris* in the tree layer at 38 years of age was significantly higher than that at 16 years of age, indicating that age had a great impact on carbon content.

**3.2.2 Carbon density of vegetation in the two types of plantation forests under different lithologies.**   As shown in Table 6, the carbon density of vegetation in *B. cinnamon* forests of the same age was in the order of dolomite>limestone>dolomitic sandstone. Meanwhile, in *C.*

**Table 3. Vegetation biomass(M±SD) composition of the two plantation species under different lithologies.**

| Lithology | Forest structure | Herbaceous layer biomass (t·hm⁻²) | | | Shrub layer biomass (t·hm⁻²) | | | Tree layer biomass (t·hm⁻²) | | | Vegetation biomass (t·hm⁻²) | | |
|---|---|---|---|---|---|---|---|---|---|---|---|---|---|
| | | Aboveground | Belowground | Subtotal | Aboveground | Belowground | Subtotal | Aboveground | Belowground | Subtotal | Aboveground | Belowground | total |
| Dolomite | *Bodinieri cinnamon* | 0.68±0.11a | 0.45±0.05af | 1.13±0.16ab | 2.08±0.20ab | 1.04±0.10a | 3.12±0.30ac | 18.39±0.46a | 5.72±0.09a | 24.11±0.55a | 21.15±0.74a | 7.21±0.22a | 28.36±0.96a |
| | *Cupressus funebris* | 0.76±0.01ab | 0.54±0.02b | 1.30±0.02be | 2.43±0.13bf | 1.12±0.04a | 3.55±0.17ad | 12.41±0.45b | 3.3±0.32b | 15.71±0.77b | 15.61±0.58b | 4.96±0.35b | 20.57±0.93b |
| Dolomitic sandstone | *Bodinieri cinnamon* | 0.98±0.13c | 0.60±0.09bc | 1.58±0.22cf | 3.03±0.19c | 1.37±0.04b | 4.4±0.22b | 4.56±0.29c | 1.44±0.06c | 6.00±0.35c | 8.57±0.35c | 3.41±0.03c | 11.98±0.36c |
| | *Cupressus funebris* | 0.65±0.04a | 0.35±0.03d | 1.00±0.06ad | 1.97±0.03ad | 0.86±0.02c | 2.83±0.05c | 21.51±0.77d | 7.26±0.47d | 28.77±1.23d | 24.13±0.76d | 8.47±0.46d | 32.60±1.20d |
| Limestone | *Bodinieri cinnamon* | 0.93±0.05cd | 0.56±0.02be | 1.49±0.07ef | 2.24±0.10bed | 1.47±0.09b | 3.71±0.19d | 9.65±0.56e | 3.12±0.08be | 12.77±0.64e | 12.82±0.41e | 5.14±0.03be | 17.96±0.44e |
| | *Cupressus funebris* | 0.83±0.02bd | 0.53±0.02bf | 1.36±0.04ef | 2.47±0.38ef | 1.13±0.10a | 3.60±0.48ad | 14.56±0.31f | 4.69±0.27f | 19.25±0.58f | 17.86±0.67f | 6.35±0.35f | 24.21±1.02f |

Note: Different lowercase letters in the same column indicate significant differences; this is applicable for all tables below.

**Table 4. Ratio of aboveground biomass to belowground biomass of the two plantation species under different lithologies.**

| Lithology | Forest structure | Herbaceous layer | Shrub layer | Tree layer | Vegetation |
|---|---|---|---|---|---|
| Dolomite | *Bodinieri cinnamon* | 1.53±0.09ad | 2.01±0.07a | 3.21±0.04ac | 2.93±0.01a |
| | *Cupressus funebris* | 1.42±0.09a | 2.17±0.05ab | 3.78±0.22b | 3.15±0.12b |
| Dolomitic sandstone | *Bodinieri cinnamon* | 1.65±0.06b | 2.20±0.08b | 3.17±0.10ac | 2.52±0.10c |
| | *Cupressus funebris* | 1.89±0.09c | 2.29±0.03b | 2.97±0.10a | 2.85±0.08a |
| Limestone | *Bodinieri cinnamon* | 1.66±0.04b | 1.53±0.04c | 3.09±0.10d | 2.49±0.09c |
| | *Cupressus funebris* | 1.56±0.01bd | 2.18±0.19b | 3.11±0.11cb | 2.81±0.05a |

*funebris* forests of the same age, the carbon density of vegetation was higher in limestone than in dolomite, indicating that lithology had a greater influence on the vegetation carbon pool in this forest. In *C. funebris* forests of different ages, carbon density was higher in older forests, indicating that age has a greater impact on vegetation carbon sink. The carbon density was higher in the tree layer, followed by the shrub and herb layers.

**3.3 Characteristics of the carbon pool of the litterfall accumulation in the two plantation forests under different lithologies in karst areas.** As shown in Table 7, the biomass of litterfall in *B. cinnamon* forest was in the order of dolomite>limestone>dolomitic sandstone. Meanwhile, in *C. funebris* forest, the biomass of litterfall was higher in limestone and dolomitic sandstone than in dolomite. The carbon content of litterfall was higher in limestone and lower in dolomite and dolomitic sandstone. The carbon density was the highest in dolomitic *B. cinnamon* forest and the lowest in dolomitic *C. funebris* forest. Therefore, different lithologies significantly affected the litterfall accumulation and its carbon density.

## 3.4 Characteristics of the soil organic carbon pool in the bare ground under different lithologies in karst region

**3.4.1 Soil volume weight, gravel content, volume, and organic carbon content of bare ground under different lithologies in karst region.** As shown in Table 8, the soil volume and gravel content of bare ground tended to increase with the deepening of soil layer. In contrast, the soil volume and soil organic carbon content displayed a decreasing trend. The soil volume weight of bare ground was high in dolomite, moderate in dolomitic sandstone, and

**Table 5. Carbon content of vegetation in the two plantation forests under different lithologies.**

| Lithology | Forest structure | Herbaceous layer carbon content (%) | | Carbon content of shrub layer (%) | | Carbon content of tree layer (%) | | | | Weighted average carbon content (%) |
|---|---|---|---|---|---|---|---|---|---|---|
| | | Aboveground | Belowground | Aboveground | Belowground | Leaf | branch | Stem | Roots | |
| Dolomite | *Bodinieri cinnamon* | 39.40±0.64a | 38.90±0.59a | 46.11±0.69ab | 45.22±1.05a | 46.40 ±0.40a | 47.41 ±0.40ab | 49.23 ±0.92a | 44.56 ±0.47a | 46.94±0.14a |
| | *Cupressus funebris* | 38.46±0.44bc | 36.18±0.62b | 45.25±0.93b | 45.26±0.82a | 48.83 ±0.85b | 46.67 ±0.82a | 49.55 ±0.90a | 45.52 ±0.44a | 46.88±0.10ab |
| Dolomitic sandstone | *Bodinieri cinnamon* | 38.71±0.32ab | 38.25±025ac | 45.35±0.60b | 45.34±0.43a | 47.28 ±0.56a | 48.71 ±1.25b | 49.22 ±0.94a | 44.44 ±0.53a | 45.65±0.77c |
| | *Cupressus funebris* | 37.71±0.42c | 37.51±0.52c | 47.11±0.29ac | 45.20±0.15a | 48.59 ±0.52b | 47.51 ±0.47ab | 51.22 ±0.87b | 48.24 ±0.96b | 48.76±0.04d |
| Limestone | *Bodinieri cinnamon* | 39.46±0.49a | 39.04±0.75a | 46.20±0.28ab | 44.64±0.52a | 46.18 ±0.81a | 47.84 ±0.74ab | 48.46 ±1.15a | 45.65 ±1.05a | 46.24±0.25bc |
| | *Cupressus funebris* | 39.56±0.51a | 38.84±0.36a | 46.50±0.48a | 45.59±0.38a | 48.72 ±0.60b | 44.88 ±0.83c | 48.61 ±0.72a | 45.26 ±0.86a | 46.60±0.38ab |

Table 6. Carbon density of vegetation in the two plantation forests under different lithologies.

| Lithology | Forest structure | Herbaceous layer carbon density (t·hm⁻²) | | | Shrub layer carbon density (t·hm⁻²) | | | Carbon density of tree layer (t·hm⁻²) | | | Vegetation carbon density (t·hm⁻²) | | |
|---|---|---|---|---|---|---|---|---|---|---|---|---|---|
| | | Aboveground | Belowground | Subtotal | Aboveground | Belowground | Subtotal | Aboveground | Belowground | Subtotal | Aboveground | Belowground | total |
| Dolomite | *Bodinieri cinnamon* | 0.27±0.04a | 0.17±0.02a | 0.44 ±0.06ad | 0.96±0.11a | 0.47±0.06a | 1.43 ±0.16ac | 9.41±0.16a | 2.03±0.13a | 11.44 ±0.22a | 10.64±0.31a | 2.68±0.15a | 13.32 ±0.42a |
| | *Cupressus funebris* | 0.29±0.01ac | 0.19±0.01ad | 0.48 ±0.01ab | 1.10±0.07ac | 0.51±0.02a | 1.61 ±0.09ad | 6.06±0.25b | 1.49±0.09b | 7.55 ±0.33b | 7.45±0.32b | 2.19±0.12b | 9.64 ±0.42b |
| Dolomitic sandstone | *Bodinieri cinnamon* | 0.38±0.05b | 0.23±0.04b | 0.61 ±0.09c | 1.37±0.10b | 0.62±0.02b | 1.99 ±0.13b | 2.24±0.14c | 0.62±0.08c | 2.86 ±0.22c | 3.99±0.18c | 1.47±0.07c | 5.46 ±0.26c |
| | *Cupressus funebris* | 0.25±0.02a | 0.13±0.01c | 0.38 ±0.03d | 0.93±0.02ad | 0.39±0.01c | 1.32 ±0.03c | 11.26±0.54d | 2.94±0.06d | 14.20 ±0.58d | 12.43±0.54d | 3.46±0.05b | 15.90 ±0.57d |
| Limestone | *Bodinieri cinnamon* | 0.37±0.03b | 0.22±0.01bd | 0.59 ±0.04c | 1.04±0.04cd | 0.65±0.05b | 1.69 ±0.09d | 4.38±0.33e | 1.65±0.05b | 6.03 ±0.34e | 5.79±0.27e | 2.52±0.08ad | 8.31 ±0.21e |
| | *Cupressus funebris* | 0.33±0.01bc | 0.21±0.01ab | 0.54 ±0.01bc | 1.15±0.17c | 0.51±0.05a | 1.66 ±0.21ad | 7.48±0.19f | 1.62±0.18b | 9.10 ±0.24f | 8.95±0.29f | 2.34±0.20d | 11.29 ±43f |

**Table 7. Carbon pool characteristics of the litterfall accumulation in the two plantation species under different lithologies.**

| Lithology | Forest structure | the litterfall accumulation (t) | Carbon content (%) | Carbon density (t·hm$^{-2}$) |
|---|---|---|---|---|
| Dolomite | *Bodinieri cinnamon* | 3.27±0.29a | 39.21±0.87a | 1.29±0.14a |
| | *Cupressus funebris* | 1.15±0.26b | 38.33±0.64a | 0.44±0.09b |
| Dolomitic sandstone | *Bodinieri cinnamon* | 1.86±0.11c | 39.73±0.67a | 0.74±0.03cf |
| | *Cupressus funebris* | 1.34±0.04bd | 39.48±1.15a | 0.53±0.00d |
| Limestone | *Bodinieri cinnamon* | 1.93±0.11ce | 42.21±1.28b | 0.81±0.02ce |
| | *Cupressus funebris* | 1.43±0.06bf | 43.81±0.52c | 0.62±0.02df |

low in limestone. The gravel content of bare ground was high in dolomitic sandstone, moderate in limestone, and low in dolomite. The soil volume of bare ground was high in limestone and dolomitic sandstone and low in dolomite. The soil organic carbon content of bare ground was high in dolomite, moderate in limestone, and low in dolomite. Soil organic carbon content in the 0–20 cm layer was high in dolomite, followed by limestone, and low in dolomitic sandstone. Therefore, soil quality varies with lithology.

**3.4.2 Carbon storage and soil organic carbon density of bare ground under different lithologies in karst.** As shown in Table 9, the carbon storage and density in the 0–20 cm soil layer of the bare ground were high in dolomite, moderate in limestone, and low in dolomitic sandstone. The carbon storage and density in the >20 cm soil layer were high in limestone, moderate in dolomite, and low in dolomitic sandstone. The carbon storage and density in the two layers together were consistent with the variation in the 0–20 cm soil layer. The carbon storage and density in the 0–20 cm soil layer were higher than those in the >20 cm soil layer. Therefore, soil carbon exhibits a surface clustering feature. Overall, carbon storage and density of dolomitic sandstone were lower, indicating the poor carbon sequestration capacity of soil developed from this parent material.

## 3.5. Characteristics of the soil organic carbon pools in the two plantation forests under different lithologies in karst region

**3.5.1 Soil gravel content, volume, and organic carbon content of 2 plantation forests under different lithologies.** As shown in Table 10, the soil organic carbon content and volume were higher in the 0–20 cm soil layer than that in the >20 cm soil layer under the three

**Table 8. Soil volume weight, gravel content, volume, and organic carbon content of bare ground under different lithologies.**

| Lithology | Volume weight (g·cm$^{-3}$) | | Gravel content (%) | | Soil volume (m$^3$) | | Soil organic carbon content (g·kg$^{-1}$) | |
|---|---|---|---|---|---|---|---|---|
| | 0–20 cm | >20 cm | 0–20 cm | >20 cm | 0–20 cm | >20 cm | 0–20 cm | >20 cm |
| Dolomite | 1.04 ±0.02a | 1.06 ±0.05a | 22.54 ±0.38a | 23.65 ±0.46a | 17.76 ±0.31a | 7.01 ±0.03a | 39.88 ±0.14a | 26.97 ±0.27a |
| Dolomitic sandstone | 0.94 ±0.05b | 1.05 ±0.03a | 28.73 ±0.52b | 30.67 ±0.57b | 19.49 ±0.42b | 7.87 ±0.31b | 32.37 ±0.33b | 23.23 ±0.48b |
| Limestone | 0.93 ±0.05bc | 1.04 ±0.02a | 25.90 ±0.8c | 26.88 ±0.93c | 20.01 ±0.35bc | 7.65 ±0.47ab | 37.31 ±0.59c | 31.12 ±0.90c |

**Table 9. Carbon storage (400 m$^2$) and organic carbon density in bare ground soils under different lithologies.**

| Lithology | Carbon storage (t) | | Total | Carbon density (t·hm$^{-2}$) | | Total |
|---|---|---|---|---|---|---|
| | 0–20 cm | >20 cm | | 0–20 cm | >20 cm | |
| Dolomite | 0.57±0.02a | 0.15±0.01a | 0.73±0.02a | 1.43±0.05a | 0.38±0.02a | 1.81±0.04a |
| Dolomitic sandstone | 0.42±0.04b | 0.13±0.00b | 0.56±0.04b | 1.06±0.09b | 0.33±0.00b | 1.39±0.09b |
| Limestone | 0.51±0.04a | 0.18±0.01c | 0.69±0.04a | 1.28±0.11a | 0.45±0.04c | 1.73±0.11a |

lithologies of the two plantations, while soil volume weight and gravel content followed the opposite trends. Organic carbon content was generally higher in the *C. funebris* forest than in the *B. cinnamon* forest under the same lithology. Organic carbon content of the 0–20 cm soil layer was higher in dolomite and limestone and lower in dolomitic sandstone. Organic matter in this layer is favorable for seedlings at the early stages of artificial vegetation restoration in dolomitic and limestone habitats. The overall soil volume was slightly higher in the *B. cinnamon* forest than in the *C. funebris* forest under the same lithology, while gravel content followed the opposite trend.

**3.5.2 Storage and density of soil organic carbon in the two plantation species under different lithologies.** As shown in Table 11, the storage and density soil organic carbon of each stand were higher in limestone, followed by dolomite, and lower in dolomitic sandstone. The storage and density of soil organic carbon in the *C. funebris* forest were higher than those in the *B. cinnamon* forest on dolomitic sandstone and limestone. No evident difference in the storage and density of soil organic carbon between the two vegetations was observed on dolomite. Therefore, dolomitic sandstone and limestone are more favorable for *C. funebris* growth. Under the same lithology, the storage and density of soil organic carbon in the 0–20 cm soil layer were significantly higher than those in the >20 cm soil layer, further indicating that soil organic carbon shows a surface clustering feature.

## 3.6 Carbon density and carbon sequestration rate of the two plantation forest ecosystems under different lithologies in karst regions

As shown in Table 12, under the three lithologies, among the components of ecosystem, carbon density was high in vegetation, followed by soil, and low in litterfall. Vegetation carbon

**Table 10. Soil volume weight, gravel content, volume, and organic carbon content of the two plantations under different lithologies.**

| Lithology | Forest | Volume weight (g·cm$^{-3}$) | | Gravel content (%) | | Soil volume (m$^3$) | | Soil organic carbon content (g·kg$^{-1}$) (g·kg$^{-1}$) | |
|---|---|---|---|---|---|---|---|---|---|
| | | 0–20 cm | >20 cm | 0–20 cm | >20 cm | 0–20 cm | >20 cm | 0–20 cm | >20 cm |
| Dolomite | *Bodinieri cinnamon* | 1.02 ±0.02a | 1.09 ±0.01a | 20.76 ±0.37a | 23.55 ±0.47a | 19.45 ±0.49ab | 7.42 ±0.43a | 40.79 ±0.83a | 28.57 ±1.04a |
| | *Cupressus funebris* | 0.86 ±0.02b | 0.91 ±0.02b | 21.44 ±0.43b | 25.44 ±0.47b | 18.45 ±0.44b | 6.08 ±0.22b | 58.65 ±0.78b | 18.04 ±0.16b |
| Dolomitic sandstone | *Bodinieri cinnamon* | 0.90 ±0.01c | 1.01 ±0.01c | 27.73 ±0.27c | 29.33 ±0.34c | 21.44 ±0.49ce | 8.40 ±0.18c | 33.88 ±0.65c | 24.38 ±0.63c |
| | *Cupressus funebris* | 0.94 ±0.01d | 1.03 ±0.01d | 28.43 ±0.40d | 29.70 ±0.33c | 20.22 ±0.55ad | 8.40 ±0.49c | 37.93 ±0.98d | 31.11 ±0.61d |
| Limestone | *Bodinieri cinnamon* | 0.96 ±0.01d | 1.02 ±0.01cd | 24.43 ±0.27e | 24.92 ±0.06b | 21.75 ±0.71ef | 9.44 ±0.46d | 37.82 ±0.43d | 33.16 ±0.44e |
| | *Cupressus funebris* | 0.87 ±0.01b | 0.96 ±0.02e | 23.39 ±0.38f | 25.50 ±0.45b | 21.30 ±0.71cf | 8.59 ±0.50c | 60.00 ±1.77b | 57.85 ±0.59f |

**Table 11. Storage (400 m²) and density of soil organic carbon in the two plantation species under different lithologies.**

| Lithology | Forest structure | Carbon storage (t) | | | Carbon density (t·hm⁻²) | | |
|---|---|---|---|---|---|---|---|
| | | 0–20 cm | >20 cm | Total | 0–20 cm | >20 cm | Total |
| Dolomite | *Bodinieri cinnamon* | 0.64 ±0.04a | 0.18 ±0.01a | 0.82 ±0.04a | 1.60 ±0.09a | 0.44 ±0.03a | 2.05 ±0.11a |
| | *Cupressus funebris* | 0.73 ±0.03b | 0.07 ±0.00b | 0.81 ±0.03a | 1.83 ±0.07b | 0.19 ±0.01b | 2.01 ±0.08a |
| Dolomitic sandstone | *Bodinieri cinnamon* | 0.47 ±0.01c | 0.15 ±0.00c | 0.62 ±0.01b | 1.18 ±0.02c | 0.37 ±0.01c | 1.55 ±0.03b |
| | *Cupressus funebris* | 0.52 ±0.00d | 0.19 ±0.01a | 0.71 ±0.02c | 1.29 ±0.01d | 0.47 ±0.03a | 1.76 ±0.04c |
| Limestone | *Bodinieri cinnamon* | 0.60 ±0.02e | 0.24 ±0.01d | 0.84 ±0.02a | 1.49 ±0.04e | 0.60 ±0.02d | 2.09 ±0.06a |
| | *Cupressus funebris* | 0.85±0.02f | 0.36 ±0.02e | 1.20 ±0.03d | 2.12±0.04f | 0.89 ±0.05e | 3.01 ±0.08d |

sink occupied the dominate position, indicating that vegetation is the most effective or contributes the most to the carbon sink in karst forest restoration. The net carbon density of the ecosystem in *B. cinnamon* forest of the same or similar age was high in dolomite, followed by limestone, and low in dolomitic sandstone. Meanwhile, in *C. funebris* of the same age, the net carbon density of the ecosystem was higher in limestone and lower in dolomitic sandstone. Therefore, dolomitic sandstone plantation forests have a poor carbon sequestration capacity. The carbon sequestration rate of the vegetation and ecosystem was high in dolomite, moderate in limestone, and low in dolomitic sandstone in *B. cinnamon* forest, whereas high in limestone, moderate in dolomite, and low in dolomitic sandstone in *C. funebris* forest. These trends support the poor carbon sequestration capacity of dolomitic sandstone plantation forests. Furthermore, the carbon sequestration capacity of plantation forest ecosystems differed with lithology, species, and plantation age, reflecting the complexity and diversity of karst lithological habitats and the difficulty of ecological restoration and carbon sink assessment in these regions.

**Table 12. Carbon density and carbon sequestration rate in the two plantation forest ecosystems under different lithologies.**

| Lithology | Forest structure | Carbon density (t·hm⁻²) | | | | Ecosystem net carbon intensity (t·hm⁻²) | Vegetation carbon sequestration rate (t·hm⁻²·a⁻¹) | Ecosystem carbon sequestration rate (t·hm⁻²·a⁻¹) |
|---|---|---|---|---|---|---|---|---|
| | | Litterfall | Vegetation | Soil | Ecosystem | | | |
| Dolomite | *Bodinieri cinnamon* | 1.29 ±0.14a | 13.31 ±0.42a | 2.05 ±0.11ae | 16.65 ±0.65a | 14.83 ±0.63a | 0.83±0.14a | 0.93±0.04a |
| | *Cupressus funebris* | 0.44 ±0.09b | 9.64±0.42b | 2.02 ±0.08ad | 12.10 ±0.44b | 10.28 ±0.42b | 0.60±0.14b | 0.64±0.03b |
| Dolomitic sandstone | *Bodinieri cinnamon* | 0.74 ±0.03cf | 5.47±0.25c | 1.55 ±0.03b | 7.76±0.31c | 6.36±0.38c | 0.42±0.02c | 0.49±0.03c |
| | *Cupressus funebris* | 0.53 ±0.00bd | 15.89 ±0.57d | 1.76 ±0.04c | 18.18 ±0.61d | 16.79 ±0.69d | 0.42±0.02c | 0.44±0.02cd |
| Limestone | *Bodinieri cinnamon* | 0.81 ±0.02e | 8.31±0.21e | 2.09 ±0.06d | 11.21 ±0.25e | 9.47 ±0.28be | 0.52±0.01d | 0.59±0.02e |
| | *Cupressus funebris* | 0.62 ±0.02df | 11.29 ±0.43f | 3.01 ±0.08e | 14.92 ±0.42f | 13.19 ±0.38f | 0.71±0.03e | 0.82±0.02f |

## 4 Discussion

### 4.1 Vegetation biomass characteristics of the two plantation forest ecosystems under different lithologies in karst region

The biomass of herbs, shrubs, and standard trees was determined using the harvesting method. However, the belowground biomass of trees was difficult to obtain because of the thin soil layer in the karst region, continuous distribution of rock in the belowground habitat, and general distribution of roots in rock pores and fractures. To overcome the difficulties of harvesting and to achieve accurate biomass determination, the present study used regression simulations of standard trees biomass in the harvested sample plots, as described in previous studies in the same area [10, 27–30]. We used the developed regression equation to determine the biomass of the tree layers. To verify the accuracy of the regression equation, the biomass of the tree layer was measured using the average standard tree method [24] and compared with the value obtained using the regression method. The results were consistent between the two methods, indicating the reliability of results obtained in the present study. Therefore, to measure the biomass of *B. cinnamon* and *C. funebris* in the same or similar study areas, the regression equations obtained in the present study can be used for accurate biomass determination based on surface dendrochronological indicators (i.e., DBH and H); this information can provide insights into vegetation biomass, particularly under different lithologies.

Native forests in karst regions have a low biomass [10, 27]. The biomass of plantation forests estimated in the present study was generally consistent with the reported values. For instance, the biomass of tree layer in *B. cinnamon* forests at 13–16 years of age ranged from 6.00 to 24.11 t·hm$^{-2}$, which is consistent with the values for artificial camphor forests at 16 years of age (tree layer: 15.18 t·hm$^{-2}$, vegetation: 17.21 t·hm$^{-2}$) in Guiyang in the same karst region [9]. The biomass of the 16-year-old *B. cinnamon* plantation forest in the present study was also similar to that of the native karst shrubland (34.28 t·hm$^{-2}$) [10, 34] over the same restoration time but was lower than that of the 16-year-old plantation forest of camphor (9.24–24.91 t·hm$^{-2}$) within the same plant family and in the same latitude-normative landform [35]. Furthermore, the biomass of 13–16-year-old *B. cinnamon* vegetation in the present study (11.98–28.36 t·hm$^{-2}$) was much lower than that of the 18-year-old plantation forest of camphor within the same family and in the same latitude-normative landform (111.08 t·hm$^{-2}$) [36]. These results indicate that plantation forests in karst regions produce lower than naturally restored native forests and plantation forests in a normal landform.

As shown in Table 13, the ratio of biomass of each tree part to the total biomass of the tree layer was in the order of stem (31.51%–41.63%) > branch (18.53%–30.06%) > root (18.52%–28.32%) > leaf (10.11%–20.59%). Therefore, stem and branches contributed the most to the overall biomass, although the contribution of roots cannot be ignored. The proportion of leaves was higher in *C. funebris* (16.40%–20.59%) than in *B. cinnamon* (10.11%–13.92%), while the branches followed the opposite trend. The proportion of leaves in *B. cinnamon* was higher

**Table 13. Biomass of organs in the tree layer as a percentage of total biomass (%).**

| Lithology | Species | Leaf | Branch | Stem | Root |
|---|---|---|---|---|---|
| Dolomite | *Bodinieri cinnamon* | 13.92 | 27.49 | 39.65 | 18.94 |
| | *Cupressus funebris* | 20.59 | 18.53 | 40.09 | 20.79 |
| Dolomitic sandstone | *Bodinieri cinnamon* | 12.76 | 24.06 | 39.94 | 23.24 |
| | *Cupressus funebris* | 16.40 | 21.12 | 41.29 | 21.19 |
| Limestone | *Bodinieri cinnamon* | 10.11 | 30.06 | 31.51 | 28.32 |
| | *Cupressus funebris* | 19.49 | 20.36 | 41.63 | 18.52 |

than that in 18-year-old camphor within the same family and genus and in a normal landform at the same latitude (4.70%) [36]. Furthermore, these values were higher than the proportion of leaves in 7-year-old camphor (7.24%-10.78%) in a normal landform at the same latitude [35] but similar to that in 16-year-old camphor (13.39%) in Guiyang within the same karst region [9]. Moreover, the weight ratio of *B. cinnamon* and *C. funebris* leaves was consistent with the ratio of leaf biomass to total biomass in the same karst region (12.28% for rowan, 8.24% for pepper, 43.04% for cypress, and 23.89% for Dodonaea viscosa) [5], suggesting higher leaf weight ratio in karst regions. However, the generalizability of this conclusion must be tested using various methods in future research. The ratio of aboveground biomass to below-ground biomass reflects the growth strategy of plants, and it is to one of the core aspects of plant life history response theory [37]. The ratio of aboveground biomass (including leaves and stems) to belowground biomass of native plants is higher in karst region than that in nor-mal landscapes [10], and our results are consistent with this conclusion. Here, the ratio of aboveground biomass (including leaves and stems) to belowground biomass ranged from 1.42 to 1.89 for the herb layer, 1.53 to 2.29 for the shrub layer, 2.97 to 3.78 for the tree layer, and 2.52 to 3.15 for the total vegetation. However, these values are lower than the global forest stem biomass-to-root biomass ratio of 3.13 to 5.88 and the Chinese forest stem biomass-to-root bio-mass ratio of 2.78 to 5.88 [38], indicating that the root biomass of karst plantation forest eco-systems accounts for a larger proportion of the overall biomass [10]. In addition, to adapt to the harsh ecological environment, plants in karst regions adopt an adaptive strategy of expand-ing root biomass to obtain more nutrients and groundwater.

## 4.2 Carbon sequestration characteristics of the two plantation forest ecosystems under different lithologies in karst regions and insights

In the present study, the carbon density in the litterfall accumulation was 0.44 to 1.29 t·hm$^{-2}$, being lower than the values from the herb stages to climax stages layers (1.53 to 4.97 t·hm$^{-2}$) in naturally restored karst forests in the same region [18]. Therefore, carbon storage in the litter-fall accumulation of an artificially restored karst forest was much lower than that of a naturally restored forest. The organic carbon content of the litterfall accumulation was 38.33% to 43.81%, which was comparable to or slightly higher than that of the native forest litterfall (37.78%–39.93%). Therefore, litterfall biomass in plantation forests (1.15–3.27 t·hm$^{-2}$) is lower than that in native forests (3.83–13.16 t·hm$^{-2}$) due to the simple species sources of litterfall in the former. In plantation forests, understory grasses and shrubs are artificially removed during the nurturing process, while in native forests, litterfall has abundant species and stable under-story grasses and shrubs. Therefore, in karst regions, due to the harshness of the surface ecol-ogy, should the grass–shrub system in the understory of the dominant species be removed entirely or retained partially at the nurturing stage to create good hydrothermal conditions in the understory when plantation forests are cultivated? How much understory should be retained? It is worthwhile to study these issues in depth in future.

To objectively reflect the characteristics of the soil organic carbon pool [10], soil organic carbon content, soil volume weight, gravel content, and volume were actually measured. The organic carbon content of the 0–20 cm soil layer in the studied karst forest (32.37–39.88 g·kg$^{-1}$) was similar to that reported in the Maolan Nature Reserve (20.89–203.5 g·kg$^{-1}$) range [12] but higher than that reported in the Huajiang River (25˚38′N, 105˚38′E) [39] (12.68 g·kg$^{-1}$ in grassland, 18.06 g·kg$^{-1}$ in shrubland, and 20.96 g·kg$^{-1}$ in forest) as well as the minimum content reported in the karst region of Southwest China (7.3 g·kg$^{-1}$) [40]. Therefore, soil organic car-bon content in karst regions is more variable. Hence, to assess soil organic carbon storage in karst regions, the actual measurement method is appropriate; otherwise, the measured results

may be inaccurate. Therefore, the actual measurement has important methodological significance in the assessment of ecosystem carbon in karst regions. Further, the density of soil organic carbon in the present study ranged from 1.55 to 3.01 t·hm$^{-2}$, being lower than that reported in naturally restored forests of Mao Lan (8.26–18.80 t·hm$^{-2}$) [12] and much lower than that of the camphor plantations of the same age in Guiyang (115.37 t·hm$^{-2}$) [9]. Therefore, the soil organic carbon content in restored plantation forests in areas with intense rocky desertification is low. Simultaneously, compared with that in the bare ground, the soil carbon density in forests only increased by 0.16 to 1.28 t·hm$^{-2}$, indicating no significant increase under each lithology. Therefore, the plantation soil has a poor carbon sequestration capacity due to low soil volume and high gravel content in areas with intense rocky desertification in karst regions. Furthermore, the lower volume of soil must provide nutrients to plants, leading to higher soil respiration and organic carbon activity in forests of karst regions than in those in forests of normal landscapes [12]. Hence, soil carbon sequestration is poor. These factors reflect the limitations and unique features of karst plantation forests to increase carbon sinks.

The aboveground carbon content of the herbaceous layer in the present study ranged from 37.71% to 39.56%, being higher than that in naturally restored forests (37.13%) [41]. The carbon content of the shrub layer in the present study ranged from 44.64% to 47.11%, being close to that in natural forests (45.93%) [41]. The carbon content of various parts of the tree layer in the present study ranged from 44.44% to 51.22%, being different from the value commonly used in large-scale studies (45% or 50%). Therefore, the carbon content of plants in karst areas is highly variable. Furthermore, carbon content is one of the important factors affecting plant carbon storage; thus, a fixed carbon content should not be used in the estimation of vegetation carbon storage in karst regions, and the actual measurement method should be adopted instead. In the present study, the vegetation carbon density in young *B. cinnamon* forests ranged from 5.47 to 13.31 t·hm$^{-2}$, being consistent with that in artificial camphor forests of same age in the same area (9.98 t·hm$^{-2}$) [9]. The vegetation carbon density in young and middle-aged *C. funebris* forests in the present study was 9.64–11.29 t·hm$^{-2}$ and 15.89 t·hm$^{-2}$, respectively. In a previous study [42], the average carbon density in young and middle-aged *C. funebris* forests in Guanling, Zhenfeng, and Huajiang in the same karst area was respectively 17.34 and 22.00 t·hm$^{-2}$; these value are higher than those obtained in the present study. This may be because before restoration, the vegetation sample plots in the present study were in an intensely rocky desertified habitat, where plant growth is difficult. However, the carbon density of vegetation in plantation forests in the present study and that reported in the study by Ma (2020) [42] were lower than the values in naturally restored shrublands (25.60 t·hm$^{-2}$) of the same age [10, 34]. Therefore, habitat, tree species, forest age, and restoration methods in karst regions affect vegetation carbon density. In addition, the carbon density of forest vegetation in karst regions shows extremely spatial variability, leading to complexity and uncertainty in the assessment of carbon pools in these ecosystems. Ecosystem carbon density constitutes the carbon density of litterfall, soil, and vegetation. The ratio of vegetation carbon density to ecosystem carbon density was 79.94% and 79.65% for dolomitic *B. cinnamon* and *C. funebris* forests, 70.49% and 87.40% for dolomitic sandstone *B. cinnamon* and *C. funebris* forests, and 74.13% and 75.67% for limestone *B. cinnamon* and *C. funebris* forests, respectively. Therefore, the ecosystem carbon pool of artificially restored plantations in karst areas with intense rocky desertification is dominated by vegetation, as opposed to the dominance of soil in the ecosystem carbon pool in restored plantations on normal landscapes. This is mainly because normal landscapes have a large soil volume with rich organic carbon pools. Conversely, soil volume in the intensely rocky desertified areas is small, and the weathering of rocks into soil in such areas is a long process. Hence, it is difficult to achieve a significant increase in soil volume within the vegetation restoration period. The ratio of vegetation carbon storage to soil carbon storage in

the present study ranged from 3.53 to 9.03, being higher than the average global value of 0.46 [43] and Chinese average of 0.36 [33]. This further confirms the greatest contribution of vegetation and the smallest of soil to the carbon pool of forest ecosystems in karst regions. These results have important implications for the establishment of carbon sink forests in karst regions. As such, it is necessary to focus on vegetation carbon sinks and pay particular attention to the selection of species and their cultivation techniques, in addition to further research and development.

The carbon density of ecosystem in the present study ranged from 7.76 to 16.65 $t \cdot hm^{-2}$, being lower than that of evergreen and deciduous broad-leaved forests on a normal landscape in the same zone [44–46]. Therefore, plantation forests in rocky desertified areas are low-carbon ecosystems, with carbon density lower than the global average for forest vegetation (86.00 $t \cdot hm^{-2}$). In the present, the net carbon density of different lithological ecosystems was high in dolomite (14.83 $t \cdot hm^{-2}$), followed by limestone (9.47 $t \cdot hm^{-2}$) and dolomitic sandstone (6.36 $t \cdot hm^{-2}$). In B. cinnamon forests of the same age, lithology had a great influence on ecosystem carbon sink, possibly due to the differences in the adaptability of plants under various belowground lithological conditions [2]. This finding has important implications in the establishment of carbon sink forests in karst areas. Specifically, the strategy of "suitable lithologies and trees" should be adopted when selecting tree species for carbon sink forests. The carbon sequestration rate of the two plantation ecosystems in the present study (0.44–0.93 $t \cdot hm^{-2} \cdot a^{-1}$) was rather different from the previously reported values in the same region (0.08 $t \cdot hm^{-2} \cdot a^{-1}$ for cypress, 0.69 $t \cdot hm^{-2} \cdot a^{-1}$ for Dodonaea viscosa, 0.28 $t \cdot hm^{-2} \cdot a^{-1}$ for pepper, and 1.18 $t \cdot hm^{-2} \cdot a^{-1}$ for rowan) [5], indicating that lithology, species, and habitat in karst areas significantly affect ecosystem carbon sequestration. The vegetation carbon sequestration rate in the present study (0.42–0.83 $t \cdot hm^{-2} \cdot a^{-1}$) was slightly lower than that at all stages of natural recovery (0.45–1.70 $t \cdot hm^{-2} \cdot a^{-1}$) [10] but significantly lower than that of 5–14-year-old alder plantations in karst areas and natural subtropical forests in Huayuan County, Xiangxi (3.68–5.33 $t \cdot hm^{-2} \cdot a^{-1}$) [47]. Moreover, the carbon sequestration rate of tree layer in the present study (0.22–0.72 $t \cdot hm^{-2} \cdot a^{-1}$) was lower than that in a 7-year-old camphor forest (0.88–1.60 $t \cdot hm^{-2} \cdot a^{-1}$) in Jiangxi [35] (28° 50′46″N, 117°33′32″E) with the same silvicultural density (1000–2000 $plants \cdot hm^{-2}$). Therefore, the carbon sequestration capacity of karst ecosystems is lower than that of normal ecosystems due to the low biomass of vegetation and the low volume of soil in the former. In summary, although the carbon sequestration capacity of karst plantation forest soil is much lower than that of vegetation, lithology and its soil influence plant growth. Specifically, as karst regions have a small volume of soil, plant roots mostly use rocks as the growth space. Furthermore, the internal spatial morphological structure of rocks and its nutrient and water endowment patterns are different, all of which strongly affect plant physiological and ecological functions; thus, different lithologies lead to different plant growth characteristics [2], affecting the carbon sequestration capacity of vegetation. Nonetheless, the carbon density in the top community ecosystem in karst areas (81.31 $t \cdot hm^{-2}$) was higher than the average carbon density of evergreen and evergreen deciduous broad-leaved forests in China (73.68 $t \cdot hm^{-2}$) but close to the global forest vegetation carbon density (86.00 $t \cdot hm^{-2}$) [10]. Hence, karst forests can have a high carbon sequestration potential as long as they can reach the top community, and their large area produces a significant carbon sink effect of vegetation. However, the key question is how to build a carbon sink forest? The present study demonstrates that lithology is the major factor affecting vegetation carbon sink and belowground rocks and their soils have a great influence on plant growth and development. Therefore, the strategy of "suitable lithologies and trees" is of great importance to enhance carbon pools in karst areas to establish carbon sink forests. In addition, vegetation cultivation techniques tailored to different lithology should be developed.

## 5. Conclusion

Actual measurements and statistics based on soil volume, soil gravel content, plant carbon content, and biomass in karst areas are fundamental to ensure the accuracy of carbon sink assessments in forest ecosystems. In particular, fixed values of plant carbon content should not be used in carbon sink assessments. Karst plantation forests are the ecosystem with low biomass, carbon density, and carbon sequestration. To cope with the harsh karst environments and obtain more nutrients and water, plants have adopted a strategy to expand the proportion of their root and leaf biomass; thus, the ratios of plant root and leaf biomass to the total biomass are higher in karst regions than in normal landscapes. At the nurturing stage, the strategy of the removal or retention of the understory grass–shrub system in karst plantation forests should be different from that in plantation forests in normal landscapes, and these strategies warrant further research. In restored plantation forest ecosystems in intensely rocky desertified karst areas, vegetation is the major contributor to carbon sinks, and the contribution of soil is small. The carbon sequestration capacity of karst plantation forest ecosystems is not as good as that of native forests and forest ecosystems in normal landform; nonetheless, the carbon sink effect of these karst ecosystems remains significant if they eventually evolve toward being the top community. Lithology, forest age, and forest structure have a great influence on the carbon pool characteristics of karst plantation forest ecosystems. Therefore, the strategy of "suitable lithologies and trees" should be adopted during the establishment of carbon sink forests in karst regions.

## Author Contributions

**Conceptualization:** Zongsheng Huang.

**Data curation:** Yuanyuan Li, Huiwen Xiang, Yuanbo Zhang, Jun Zou, Yuhong Fu, Changjiang Qian.

**Formal analysis:** Yuhong Fu, Changjiang Qian.

**Investigation:** Yuanyuan Li, Huiwen Xiang, Yuanbo Zhang, Jun Zou, Yuhong Fu, Changjiang Qian.

**Methodology:** Yuanyuan Li, Huiwen Xiang, Yuanbo Zhang, Jun Zou, Yuhong Fu, Changjiang Qian.

**Resources:** Yuanyuan Li, Huiwen Xiang, Jun Zou.

**Supervision:** Zongsheng Huang.

**Writing – original draft:** Yuanyuan Li, Huiwen Xiang.

**Writing – review & editing:** Zongsheng Huang.

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
