## [Decision Letter · Decision Letter 0]

1 Mar 2022

PONE-D-22-03616Carbon sequestration characteristics of two plantation forest ecosystems with different lithologies of karstPLOS ONE

Dear Dr. Huang,

Thank you for submitting your manuscript to PLOS ONE. After careful consideration, we feel that it has merit but does not fully meet PLOS ONE’s publication criteria as it currently stands. Therefore, we invite you to submit a revised version of the manuscript that addresses the points raised during the review process.

We look forward to receiving your revised manuscript.

Kind regards,

Dafeng Hui, Ph.D.

Academic Editor

PLOS ONE

Journal Requirements:

Additional Editor Comments:

I now have one report from an expert reviewer who recognized the merits of the study, but also raised some concerns with the current version of the manuscript. A major revision is recommended. The authors need to address the reviewer's concerns and make a substantial revision of the manuscript.

Reviewers' comments:

Reviewer's Responses to Questions

**Comments to the Author**

1. Is the manuscript technically sound, and do the data support the conclusions?

Reviewer #1: Yes

2. Has the statistical analysis been performed appropriately and rigorously? 

Reviewer #1: Yes

3. Have the authors made all data underlying the findings in their manuscript fully available?

Reviewer #1: Yes

4. Is the manuscript presented in an intelligible fashion and written in standard English?

Reviewer #1: Yes

5. Review Comments to the Author

Reviewer #1: The study by Li et al. reports the carbon sequestration characteristics of two plantation forest ecosystems with different lithologies of karst in Guizhou, China. The manuscript is easy to follow and provides some additional information on the carbon sink assessment of forest ecosystems in karst regions. I have some minor comments and hope the authors find them helpful.

The study region: It would be helpful if the authors can show the map of karst regions in Guizhou, China and the sampling sites (if the specific coordinates are available) on the map.

Line 128-158: In the methods, I suggest the authors provide the unit of each measured variable such as biomass, carbon storage, carbon density etc. It will increase the clarity of the methods section.

The authors can also use figures to show some of the results (e.g., Table 3), which will help readers to visually check the difference among different lithologies or forests.

Line 328-331: I did not see these results (i.e. the biomass estimated versus estimated using allometric equations). Probably the authors can make a plot to show this consistency if possible.

Line 345-346: Perhaps the authors can briefly explain the difference between karst regions and other normal landforms, just to support this conclusion.

Line 359: Maybe it is also interesting to briefly discuss why leaf weight ratio is higher in karst regions.

Line 368-370: The authors need to cite some literature to support their statement here.

Line 473-474: Is this conclusion also found in other studies? Or did other studies find other important factors? I understand that there might be not many literature studying karst ecosystems, however, it will be helpful if the authors can provide more knowledge from other related studies (focusing on karst regions in other continents) so the reader will better know how this study improves our understanding of karst carbon sequestration.

6. PLOS authors have the option to publish the peer review history of their article (what does this mean?). If published, this will include your full peer review and any attached files.

Reviewer #1: No

---

## [Author Response · Author response to Decision Letter 0]

16 Sep 2022

Dear Editor,

We would like to thank the editor for giving us a chance to resubmit the paper, and also thank the reviewer for giving us constructive suggestions which would help us both in English and in depth to improve the quality of the paper. Here we submit a new version of our manuscript with the title “Carbon sequestration characteristics of two plantation forest ecosystems with different lithologies of karst” which has been modified according to the reviewer’s suggestions. Efforts were also made to correct the mistakes and improve the English of the manuscript. We mark all the changes in yellow in the revised manuscript.

Sincerely yours

The following is a point-to-point response to the reviewer’s comments.

Reviewer #1:

The study region: It would be helpful if the authors can show the map of karst regions in Guizhou, China and the sampling sites (if the specific coordinates are available) on the map.

Answer: I'm sorry, because the Chinese government is very strict in map management, it is not allowed to measure or provide topographic maps to foreign journals, so this study can only indicate the geographical coordinates of the research sample land in Table 1 of the main body.

Line 128-158: In the methods, I suggest the authors provide the unit of each measured variable such as biomass, carbon storage, carbon density etc. It will increase the clarity of the methods section.

Answer: Thank the reviewer for the comments. We have added the unit of each measured variable such as biomass, carbon storage, carbon density etc.

The authors can also use figures to show some of the results (e.g., Table 3), which will help readers to visually check the difference among different lithologies or forests.

Answer: Thank the reviewer for the comments. We have tried to use figures to show some of the results (e.g., Table 3). However, on the one hand, the graph increases the length of the paper. On the other hand, as data, the graph of the paper has no direct data, but the table can provide data intuitively. So we didn't use figures to show some of the results.

Line 328-331: I did not see these results (i.e. the biomass estimated versus estimated using allometric equations). Probably the authors can make a plot to show this consistency if possible.

Answer: Thank the reviewer for the comments. Here we use an example to illustrate this problem. As shown in Table 3 in this paper, the total biomass (tree layer biomass is 19.25 t·hm-2) of cypress funebris in limestone used allometric equations. In the limestone, we investigated three cypress funebris sample plots, each with an area of 400 square meters. The standard wood of one sample plot is shown in the table below:

Table (1) Total biomass of standard wood

Standard wood number Actual investigation DBH grouping(cm) Standard diameter at breast height(cm) Standard wood height(m) Total biomass harvested(kg)

1 2-4 3.7 3.3 12.16

2 4-6 5.5 4.5 7.23

3 6-8 7.6 4.4 3.75

As shown in Table (1), the average total biomass of each standard tree is 7.71 kg. There are 99 cypress funebris trees in the sample plot of 400 square meters. The total biomass in the 400 square meter sample plot is 763.29 kg (7.71*99=763.29). Converted to biomass per hectare is 19.08 t·hm-2. There was no significant difference between 19.08 t·hm-2 and 19.25 t·hm-2. However, limited to the length of the paper, we did not make a detailed description in the paper.

Line 345-346: Perhaps the authors can briefly explain the difference between karst regions and other normal landforms, just to support this conclusion.

Answer: Thank the reviewer for the comments. This has been explained in Line 57-59 in this paper.

Line 359: Maybe it is also interesting to briefly discuss why leaf weight ratio is higher in karst regions.

Answer: Thank the reviewer for the comments. We just found this phenomenon. Whether this phenomenon is universal needs more research and practice to test. Therefore, we dare not explain the reason, and the explanation of the reason also needs experimental research.

Line 368-370: The authors need to cite some literature to support their statement here.

Answer: Thank the reviewer for the comments. The 10th reference in this paper can support their view. So we add this reference here.

Line 473-474: Is this conclusion also found in other studies? Or did other studies find other important factors? I understand that there might be not many literature studying karst ecosystems, however, it will be helpful if the authors can provide more knowledge from other related studies (focusing on karst regions in other continents) so the reader will better know how this study improves our understanding of karst carbon sequestration.

Answer: Thank the reviewer for the comments. We explain this from two aspects. On the one hand, we directly found for the first time that lithology has an impact on carbon sink of vegetation ecosystem in karst area, and there are few literatures on this aspect. On the other hand, some literatures show that lithology has an impact on the growth of plants in karst area (such as references in Article 20 and Article 21 of this paper), which indirectly shows that lithology has an impact on the carbon sink of vegetation ecosystem in karst area.

---

## [Decision Letter · Decision Letter 1]

10 Oct 2022

Carbon sequestration characteristics of two plantation forest ecosystems with different lithologies of karst

PONE-D-22-03616R1

Dear Dr. Huang,

We’re pleased to inform you that your manuscript has been judged scientifically suitable for publication and will be formally accepted for publication once it meets all outstanding technical requirements.

Kind regards,

Dafeng Hui, Ph.D.

Academic Editor

PLOS ONE

Additional Editor Comments (optional):

The authors have adequately addressed the reviewer's concerns.

Reviewers' comments:

Reviewer's Responses to Questions

**Comments to the Author**

1. If the authors have adequately addressed your comments raised in a previous round of review and you feel that this manuscript is now acceptable for publication, you may indicate that here to bypass the “Comments to the Author” section, enter your conflict of interest statement in the “Confidential to Editor” section, and submit your "Accept" recommendation.

Reviewer #1: All comments have been addressed

2. Is the manuscript technically sound, and do the data support the conclusions?

Reviewer #1: Yes

3. Has the statistical analysis been performed appropriately and rigorously? 

Reviewer #1: Yes

4. Have the authors made all data underlying the findings in their manuscript fully available?

Reviewer #1: Yes

5. Is the manuscript presented in an intelligible fashion and written in standard English?

Reviewer #1: Yes

6. Review Comments to the Author

Reviewer #1: I thank the authors for considering my previous suggestions and I have no additional comments.

Heng Huang

7. PLOS authors have the option to publish the peer review history of their article (what does this mean?). If published, this will include your full peer review and any attached files.

Reviewer #1: No

---

## [Editor Report · Acceptance letter]

9 Nov 2022

PONE-D-22-03616R1 

Carbon sequestration characteristics of two plantation forest ecosystems with different lithologies of karst 

Dear Dr. Huang:

I'm pleased to inform you that your manuscript has been deemed suitable for publication in PLOS ONE. Congratulations! Your manuscript is now with our production department. 

Kind regards, 

on behalf of

Dr. Dafeng Hui 

Academic Editor

PLOS ONE